# Enhancing Tabular Learners with Context-Aware Semantic Embeddings

**Günther Schindler** [1]   **Maximilian Schambach** [1]   **Johannes Höhne** [1]

## Abstract

While modern tabular learners excel at capturing statistical patterns, they frequently operate in a semantic vacuum, treating categorical values as discrete symbols and ignoring the rich world knowledge inherent in feature names and cell entries. We propose Context-Aware Semantic Embeddings (CASE), a novel framework that bridges the gap between the semantic understanding of Large Language Models (LLMs) and the statistical capabilities of tabular learners. Unlike existing methods that embed rows in isolation, CASE utilizes a contextualization strategy: we pre-fill a tabular LLM's KV cache with a representative sample of rows to establish a persistent anchor of the dataset's semantic distribution. This ensures that generated row embeddings are dynamically contextualized, resolving semantic ambiguities and anchoring representations in domain-specific context. Our experiments across several benchmarks (CARTE, TextTab, and TabArena) demonstrate that CASE significantly improve performance – particularly in low-data regimes and on semantically rich datasets – setting a new state of the art when combined with tabular in-context learners.

## 1. Introduction

The dominance of tabular learners such as Gradient Boosted Decision Trees (GBDTs) and recent in-context learners stems from their ability to find statistical pattern in the feature space. However, these models operate in a semantic vacuum: they rely entirely on the provided features to infer relationships, treating categorical values as discrete symbols. In contrast, LLMs possess vast world knowledge acquired during pre-training, allowing them to recognize concepts, hierarchical relationships, and domain-specific nuances that are not explicitly defined in a single table.

The challenge lies in effectively combining the concept reasoning of LLMs with the statistical reasoning of tabular learners. Conventional approaches either use heurstic featurization of string or text columns, such as TF-IDF or other n-gram features as used in AutoGLuon or the skrub library, or generate cell- or row-level embeddings in isolation (Spinaci et al., 2025; Lefebvre et al., 2025). While useful, these methods suffer from a lack of distributional semantic awareness: A single row processed in a vacuum lacks the anchor points necessary to resolve ambiguity or interpret values relative to the rest of the dataset.

We introduce Context-Aware Semantic Embeddings (CASE) to bridge this gap. By utilizing TabGemma (Schindler et al., 2025) – a recent language model optimized for tabular prediction – we employ a "Context Priming" strategy. By pre-filling the model's KV cache with a representative sample of $k$ rows, we establish a persistent semantic manifold tailored to the specific dataset. This primed state ensures that when an individual row is embedded, its representation is not merely a translation of its strings, but a contextualized vector positioned relative to the table's global semantics.

We then combine the obtained embeddings with recent tabular in-context learners to achieve both statistical and semantic grounding: thus, our approach offers a best-of-both-worlds paradigm: the LLM acts as a sophisticated feature engineer with semantic grounding, while the downstream statistical learner performs the heavy lifting of statistical pattern detection and prediction. By projecting the high-dimensional CASE into a lower dimensional space via PCA, we provide a computationally efficient way to infuse world reasoning into any tabular pipeline.

## 2. Related Work

**Tabular Learning Baselines:** Tabular prediction has historically been dominated by GBDTs (Chen & Guestrin, 2016; Ke et al., 2017; Prokhorenkova et al., 2018). While robust, these models lack cross-task transferability and require significant per-dataset optimization. Recent deep learning architectures, such as TabR, CARTE, RealMLP, or TabM (Gorishniy et al., 2024; Kim et al., 2024; Holzmüller et al., 2024; Gorishniy et al., 2025), have achieved parity with GBDTs by incorporating specialized inductive biases, though require re-training on each table.

[1]SAP SE. Correspondence to: Günther Schindler <guenther.schindler@sap.com>.

*Proceedings of the $2^{nd}$ ICML Workshop on Foundation Models for Structured Data*, Seoul, South Korea. 2026. Copyright 2026 by the author(s).

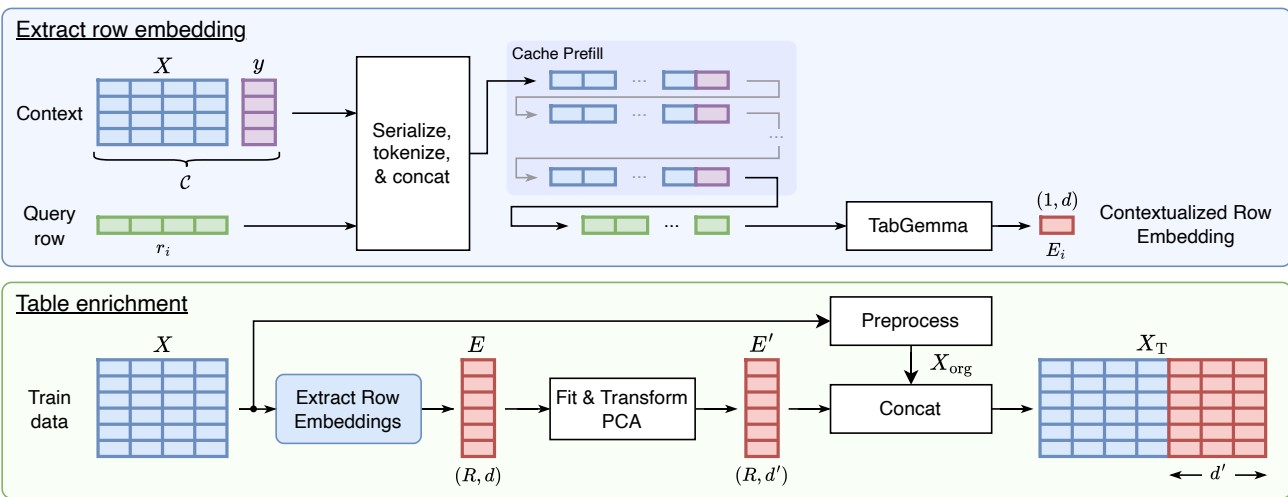

*Figure 1.* Overview of our propose contextualized row-based table enrichement pipeline. We prefill the KV cache of TabGemma using a context subsample of the training data. Rows are embedded by using the last output token of the sequence before the prediction tokens of TabGemma. For enrichment of the feature matrix $X$, individual rows are embedded, PCA-transformed and appended to the feature matrix.

**In-Context Learning (ICL):** Tabular ICL was pioneered by TabPFN (Hollmann et al., 2023), which proved transformers could perform in-context classification on small datasets. This paradigm has evolved from row-level encodings to more scalable cell-based methods like TabPFNv2, TabICL, Mitra, and ConTextTab (Hollmann et al., 2025; Qu et al., 2025; 2026; Zhang et al., 2025; Spinaci et al., 2025). While these models excel at structural pattern matching, they often treat values as abstract tokens, overlooking the latent semantic richness inherent in the data.

**LLMs and Table Semantics:** To bridge the gap between statistics and semantics, models like TabLLM, LIFT, TabuLa, and TabGemma (Hegselmann et al., 2023; Dinh et al., 2022; Gardner et al., 2024; Schindler et al., 2025) leverage the world knowledge of LLMs. These methods excel in low-data regimes where semantic priors compensate for sparse signals, though they often struggle with numeric-heavy datasets. Recent work has also explored LLM- or knowledge graph-based enrichment of conventional tabular learners (Lefebvre et al., 2025). Our work extends this by using KV-cache priming to provide a distributional anchor.

## 3. Methodology

An overview of our approach is depicted in Figure 1. Let $\mathcal{T} = \{r_1, r_2, ..., r_n\}$ be a tabular dataset with features $X$ and a target $y$. The core objective of our approach is to generate a row embedding that is aware of the global semantics of the table. This is achieved by utilizing the autoregressive nature of TabGemma and its internal state mechanisms.

**Context Priming:** To establish the semantic context of the table using TabGemma's limited context window of 128k tokens, we employ a Context Priming strategy.

(1) Sampling: We randomly sample a subset $\mathcal{C}$ from the training split of the table. (2) State Injection: The context is then serialized into a single string sequence and passed through TabGemma – an LLM pretrained on tabular prediction tasks. (3) Cache Persistence: Rather than discarding the hidden states, we retain the resulting KV cache. This cache now contains the distilled signals of the table's schema and semantic distribution, effectively acting as a permanent prefix for all subsequent computations.

**Contextualized Encoding:** With the KV cache pre-loaded, each target row $r_i \in \mathcal{T}$ is fed into the model to create a latent embedding $E_i$. Due to the pre-filled states in the KV cache, the resulting hidden representation is automatically contextualized. We extract the embedding $E_i$ from the last hidden layer of the model at the final token position of the row, capturing the integrated semantic summary of the contextualized record.

**Feature Fusion and Dimensionality Reduction:** The raw embeddings $E_i$ reside in a high-dimensional space ($d = 3840$), which can lead to problems when fed into tabular learners, e.g. memory issues or degrading performance. Therefore, we perform the following:

(1) Projection: We apply Principal Component Analysis (PCA) to the set of all contextualized embeddings $\{E_1, ..., E_n\}$, reducing them to a compact latent space of $d'$ components. (2) Concatenation: The final feature vector for the downstream learner is defined as:

$$X_{\mathrm{T}} = [X_{\mathrm{org}} \parallel \mathrm{PCA}(E))]$$

where $\parallel$ denotes the concatenation of the original preprocessed features $X_{\mathrm{org}}$ and the new semantic features.

*Table 1.* Evaluation results on the investigated benchmarks, report accuracy (Acc) and (soft-clipped) $R^2$ for classification and regression.

| Model | All Rank | All Acc | All $R^2$ | CARTE Rank | CARTE Acc | CARTE $R^2$ | TextTab Rank | TextTab Acc | TextTab $R^2$ | TabArena Rank | TabArena Acc | TabArena $R^2$ |
|---|---|---|---|---|---|---|---|---|---|---|---|---|
| **TabICL + CASE (ours)** | **2.31** | **86.5** | **76.1** | **1.24** | **81.0** | **77.5** | **2.45** | **85.7** | **68.2** | 3.33 | 88.3 | 78.5 |
| AutoGluon | 2.44 | 86.0 | 74.2 | 2.51 | 78.8 | 74.0 | 3.10 | 83.5 | 67.0 | 2.12 | 88.6 | **80.7** |
| ConTextTab | 3.54 | 85.1 | 71.2 | 3.29 | 77.1 | 72.4 | 3.35 | 84.4 | 58.8 | 3.86 | 87.6 | 77.8 |
| TabPFN | 3.84 | 84.2 | 64.8 | 5.77 | 70.7 | 59.9 | 4.00 | 81.5 | 63.8 | **1.84** | **88.7** | 80.5 |
| TabICL | 4.07 | 84.3 | 61.1 | 6.10 | 70.5 | 55.3 | 4.05 | 82.5 | 60.2 | 2.04 | **88.7** | 79.9 |
| XGBoost | 4.48 | 84.2 | 69.1 | 4.96 | 73.5 | 66.9 | 4.15 | 81.4 | 65.4 | 4.12 | 87.9 | 78.8 |
| TabGemma | 4.88 | 83.7 | 64.1 | 3.80 | 79.1 | 70.8 | 4.50 | 84.7 | 38.9 | 6.10 | 84.9 | 64.8 |
| Naive | 7.57 | 70.1 | -3.5 | 7.94 | 53.0 | -1.7 | 7.50 | 70.4 | -5.5 | 7.20 | 75.0 | -7.3 |

## 4. Experiments

**Evaluation:** We evaluate across three distinct benchmark suites: CARTE (Kim et al., 2024), TextTab (Mráz et al., 2025), and the single-fold variant of TabArena (Erickson et al., 2025). Both CARTE and TextTab are specifically designed to emphasize semantic relationships in tabular data, aligning with the core motivation of our work. Conversely, TabArena serves as a numerics-heavy baseline, included to test the robustness of our approach in environments where statistical signals dominate over semantic ones. All benchmarks cover both classification and regression tasks.

**Baselines:** We compare against a diverse set of competitive baselines, covering conventional per-dataset trained and HPO-tuned ones (XGBoost), tabular in-context learners (TabPFN, TabICL, and ConTextTab), the LLM-native TabGemma, and the AutoGluon framework (Erickson et al., 2020). Further details are provided in Appendix A.1.

### 4.1. Main Results

The main results are summarized in Table 1. For our primary implementation of CASE, we utilize a TabGemma checkpoint with a context priming window of $k = 128$ and a PCA-reduced latent space of $d' = 32$. We evaluate the synergy of CASE in combination with TabICL (v2) (Qu et al., 2026) as it is the best-performing ICL for numerics-focus tables. For preprocessing, we adhere to the recommended configurations of Skrub's `TableVectorizer` to ensure a fair comparison with established statistical pipelines.

**Performance on Semantic Benchmarks:** On the semantically rich CARTE and TextTab benchmarks, our approach significantly outperforms all existing baselines, establishing a new state-of-the-art for both classification and regression tasks, with a leading gap of 2-3 percentage points to the next-best baseline. The substantial margin of improvement over TabICL, AutoGluon, and ConTexTab suggests that for datasets where column headers and categorical values carry deep external meaning, the infusion of LLM-based world knowledge is more effective than pure statistical feature engineering or isolated LLM embeddings.

**Boundaries of Semantic Priming:** We observe a marginal performance trade-off on the TabArena benchmark, where the raw statistical learners (TabICL and AutoGluon) maintain a lead. This is expected, as TabArena is a numerics-heavy suite where the primary predictive signals are numerical rather than semantic. Importantly, this highlights the specialized nature of CASE: it acts as a modular semantic enhancer that can be selectively deployed. In practice, our results suggest a clear heuristic: CASE provide maximum utility for tables with high-cardinality text or domain-specific entities, while standard statistical learners remain optimal for purely numerical data.

### 4.2. Sample Efficiency and Few-Shot Performance

Figure 2 illustrates model performance as a function of training set size evaluating a fixed test set. To evaluate the robustness of our approach in data-constrained environments, we subsampled the CARTE benchmark datasets into training subsets ranging from 128 to 8k rows.

Our results indicate that in low-data regimes, CASE are highly dominant, outperforming state-of-the-art baselines by a significant margin. This suggests that the semantic priors embedded in the KV-primed representations provide a critical advantage when statistical signals are sparse. Furthermore, we observe a powerful synergy between the constituent models: while raw TabGemma often exhibits inconsistent performance on pure regression tasks, its high-dimensional latent features provide a rich substrate for the tabular learner. Thus, CASE successfully reconcile the semantic reasoning of LLMs with the robust numerical optimization of TabICL, effectively masking the individual weaknesses of each.

### 4.3. Ablation Study

We conduct a series of ablation experiments on the CARTE and TextTab benchmarks to validate our core hypotheses regarding the model architecture and feature representation.

**Impact of Contextualization:** To isolate the benefit of the KV-cache priming, we compare CASE ($k = 128$)

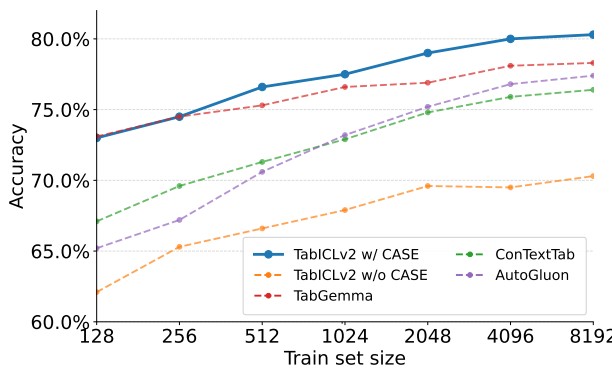

*(a)* Average Accuracy on classification.

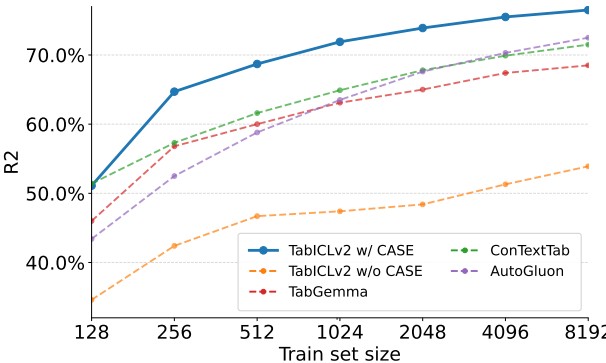

*(b)* Average R² on regression.

*Figure 2.* Impact of Context-Aware Semantic Embeddings (CASE) on predictive performance of varying few-shot degree using a fixed test set. While TabGemma struggles with regression, its latent semantic features act as a high-fidelity substrate for TabICL. This synergy allows TabICL plus CASE to achieve superior prediction quality with significantly fewer training samples than other tabular learners.

*Table 2.* Ablations on the CARTE and TextTab benchmarks.

| Model | Rank | **Acc** | **R²** |
|---|---|---|---|
| **TabICL [TF-IDF + CASE] (ours)** | **1.97** | **83.1** | **75.5** |
| TabICL [Default + CASE] (ours) | 3.38 | 82.5 | 73.9 |
| TabICL [Row-Embedding] | 3.52 | 81.7 | 73.7 |
| TabICL [Cell-Embedding] | 4.28 | 81.2 | 72.0 |
| TabICL [w/o $X_{\text{org}}$] | 4.41 | 82.1 | 71.7 |
| TabICL [CASE – Gemma3] | 4.48 | 81.8 | 72.5 |
| TabICL [TF-IDF] | 4.76 | 80.9 | 69.7 |
| TabICL [Default] | 7.06 | 75.9 | 56.4 |

against a non-contextualized row-embedding baseline (`TabICL[Row-Embedding]`, $k = 0$). The results are depitcted in Table 2. The significantly degraded performance of the $k = 0$ variant confirms that row-level semantics are inherently distributional; without the "world state" provided by the context rows, the model fails to resolve the semantic ambiguities present in the table.

**Row vs. Cell Embeddings:** We evaluate the efficacy of cell-level representations (`TabICL [Cell-Embedding]`) by replacing our row embeddings with a standard Sentence Transformer, `all-MiniLM-L6-v2`, to embed individual cells. While this yields marginal improvements over TF-IDF-based Skrub, it consistently lags behind our contextualized row embeddings. This suggests that the inter-feature relationships captured with CASE are more informative than aggregated independent cell vectors.

**Importance of Tabular-Specific LLMs:** We replaced the specialized TabGemma backbone with an off-the-shelf Gemma-3-12B model (Team et al., 2025). The significant performance drop underscores that generic LLMs, while possessing vast world knowledge, require adaptations – as shown by TabGemma – to correctly interpret the structural and relational nuances of tabular data.

**Role of Statistical Preprocessing:** Finally, we evaluate the synergy between traditional preprocessing and semantic enhancement by comparing TabICL with and without TF-IDF features using Skrub's `TableVectorizer`. Our results demonstrate that while classical statistical techniques remain a robust baseline, the addition of CASE consistently yields significant performance gains regardless of the preprocessing used. Notably, the results suggests that our framework can effectively bypass traditional feature engineering, extracting semantic signals directly from raw tabular inputs.

## 5. Conclusions

**Limitations and Future Work:** While CASE demonstrate significant potential, our current implementation has some limitations to be adressed in future works. First, as our framework is built upon TabGemma, it inherits its constraints: a lack of permutation equivariance with respect to both row and column ordering, and the limited numerical inductive bias inherent in token-based language models. This motivates further research into developing LLM architectures specifically optimized for tabular structures. Second, using PCA to project the high-dimensional embeddings into a compact latent space may be suboptimal. Future work could investigate non-linear manifold learning techniques, such as UMAP or variational autoencoders, which may better preserve the subtle semantic variances in the data.

**Summary:** We introduced Context-Aware Semantic Embeddings, designed to bridge the gap between the semantic world knowledge of LLMs and the statistical robustness of tabular learners. By utilizing a KV-cache priming strategy with TabGemma we provide a mechanism for row representations to be anchored within a table's specific distribution. Our approach achieves state-of-the-art results on semantically rich benchmarks, in particular in the low-data regime.

## Acknowledgements

We would like to thank Johannes Hoffart and Markus Kohler for their insightful comments and suggestions throughout the development of this work. We thank Myung Jun Kim for providing valuable feedback on the draft of this contribution.

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

# A. Appendix

## A.1. Baselines

**TabICL:** We use the model from the official `tabicl` package at version 2.0.3 using its default feature preprocessing if not marked otherwise specifically.

**TabPFN:** We use the model from the official `tabpfn` package at version 7.1.1, corresponding to the TabPFN-2.6 checkpoint, using its default feature preprocessing if not marked otherwise specifically.

**TabGemma:** We report TabGemma results using using the checkpoint reported by Schindler et al. (2025) with $k = 128$ retrieved context examples (at a maximum of 32k tokens).

**AutoGluon:** We evaluate AutoGluon v1.5 with its native feature encoder. We use the `extreme` preset with a per-dataset time limit of 1 h running on a 40-core node with 320 GB RAM and a single H100 GPU. (We have found the "extreme" preset to yield slightly better results than the `best_quality` at time limit of 4 h.)

**ConTextTab:** We evaluate ConTextTab v1.1.2 using the reference implementation and checkpoint[1]. We set a context size of 8k samples and evaluate with 8-fold bagging.

**XGBoost:** We use the XGBoost wrapper of `pytabkit` package (Holzmüller et al., 2024) v1.7.3. The estimator is hyperparameter-optimized and ensembled via 5-fold inner cross-validation, providing best-in-class GBDT performance. We use the hyperparameter space as define in TabArena (Erickson et al., 2025).

**Naive:** We use the Naive predictor from sklearn v1.5.2, estimating the median for regression tasks or the majority class in the case of classification tasks.

## A.2. Feature preprocessing

If available, we use the default feature preprocessing of the individual baselines investigated. For Naive and XGBoost, we use the preprocessor from AutoGluon. For TF-IDF features, we use Skrub's `TableVectorizer`, which uses a pass-through for low-cardinal and numerical features to be handled natively by TabICL, and the TF-IDF-based `StringEncoder` for high-cardinal features with more than 40 classes, using a fixed seed.

---

[1] github.com/SAP-samples/contexttab

