# OpenReview forum: "Enhancing Tabular Learners with Context-Aware Semantic Embeddings"
_ICML.cc/2026/Workshop/FMSD — FMSD @ ICML 2026 Poster_

### Official Review · Reviewer_7oBN · 2026-05-16
**Promising CASE method, but leakage concerns need clarification**

**Rating:** 6
**Confidence:** 3

**Review:**

**Summary:**

The paper proposes **CASE**, which primes TabGemma’s KV cache with sampled training rows, extracts contextualized row embeddings, reduces them with PCA, and appends them to tabular features for TabICL-style prediction. The idea is relevant and interesting: it combines LLM semantic knowledge with strong tabular learners. Results are promising on semantic benchmarks such as CARTE and TextTab, but less convincing on numerics-heavy TabArena.

**Strengths:**

timely problem, simple modular method, strong fit for the workshop, promising gains on semantic/low-data settings, and useful ablations.

**Weaknesses:**

the paper must clarify possible label leakage. Figure 1 suggests the cache context may include \(y\); the authors should state whether labels are used, whether the query row is excluded from its own context, and whether PCA is fit only on training data. Runtime/memory costs and sensitivity to \(k\) and PCA dimension should also be reported.

**Detailed comments:**

Table 1 likely has a typo: TabPFN All Acc is reported as 8.42. Claims of “significant” improvement should be supported with confidence intervals, significance tests, or per-dataset win/loss counts. The related work could also mention additional recent embedding-based table representation methods, such as **TabEmb** and **TabEmbed**, to better position CASE relative to work on LLM/tabular embeddings beyond row-level prediction.

**Justification:**

This is a promising workshop contribution, but the current version leaves important evaluation details unclear.

---

### Official Review · Reviewer_U8nQ · 2026-05-21
**A promising pre-filling strategy for tabular data with textual features.**

**Rating:** 7
**Confidence:** 4

**Review:**

Summary

This paper introduces a mechanism to create context-aware embeddings for categorical features in tabular prediction tasks. Row embeddings are generated by prefilling a Large Language Model's (LLM) Key-Value (KV) cache with a subset of the training set, anchoring the embeddings to the dataset's specific semantic distribution.

Strengths

1.	Demonstrates strong performance against tabular-only baselines (e.g., AutoGluon, Tabular Foundation Models) across multiple datasets.
2.	Offers innovative insights into the necessity of grounding textual features within a dataset's specific semantic distribution and introduces a novel prefilling strategy to achieve this.
3.	Provides solid ablation studies comparing the proposed approach against standard statistical preprocessing methods.

Areas for Improvement

1.	Main results: Some of the tested datasets, particularly TextTab, consist of complex text columns rather than simple categorical features. A stronger baseline would be the recipe introduced in TextTab; Tabular model + (text embeddings + PCA/SHAP) ideally with text embeddings generated using TabGemma. If I’m not mistaken this is (TabICL[Row-Embedding], k=0) in the ablations, this should be elevated to the main results.
2.	Computational overhead: The paper lacks a comparison of test-time compute latency and hardware requirements against other approaches.
3.	Model reliance: The framework demonstrates a strong dependency on tabular-specific, pre-trained LLMs.

Detailed Comments

1.	Have you evaluated this approach using a dedicated embedding model (e.g., Qwen3-Embedding-8B) instead of a standard LLM?

Justification of Score
 This paper demonstrates key performance improvements on tabular datasets containing textual features by grounding textual embeddings in the dataset's specific semantic distribution via a KV-cache prefilling strategy.

---

### Official Review · Reviewer_TK4D · 2026-05-22
**Review of "Enhancing Tabular Learners with Context-Aware Semantic Embeddings"**

**Rating:** 4
**Confidence:** 4

**Review:**

The computational overhead is far too high. The biggest advantage of traditional tabular models is their lightweight nature, so passing data through a large model is overly expensive. Meanwhile, after the large model finally extracts features containing complex non-linear relationships, too much information is lost during PCA dimensionality reduction. Furthermore, the performance improvement cannot be effectively attributed to either the traditional model or the LLM.